# INSTANCE-AWARE IMAGE COMPLETION

## ABSTRACT

Image completion is a task that aims to fill in the missing region of a masked image with plausible contents. However, existing image completion methods tend to fill in the missing region with the surrounding texture instead of hallucinating a visual instance that is suitable in accordance with the context of the scene. In this work, we propose a novel image completion model, dubbed *Refill*, that hallucinates the missing instance that harmonizes well with - and thus preserves - the original context. *Refill* first adopts a transformer architecture that considers the types, locations of the visible instances, and the location of the missing region. Then, *Refill* completes the missing foreground and background semantic segmentation masks within the missing region, providing pixel-level semantic and structural guidance to generate missing contents with seamless boundaries. Finally, we condition the image synthesis blocks of *Refill* using the completed segmentation mask to generate photo-realistic contents to fill out the missing region. Experimental results show the superiority of *Refill* over state-of-the-art image completion approaches on various natural images.

## 1 INTRODUCTION

Image completion is the task of restoring the masked regions in an image, which requires an understanding of the unmasked instances and the various relationships among them. Researchers have worked to develop image completion models for practical applications, such as image editing (Jo & Park, 2019; Ling et al., 2021), restoration (Wan et al., 2020; Liang et al., 2021), and object removal (Shetty et al., 2018). Most previous models, however, focus on filling in the missing region realistically without considering the instance that needs to be restored. For example, we observe that even the cutting-edge image inpainting model (Li et al., 2022) tends to complete the missing region with surrounding textures rather than attempting to restore the lost instance; this limits the usage of image completion models in real-world applications.

Removal of a focal instance in a scene can lead to substantial context change. For example, the removal of the horse in the image of Figure 1 changes the local context around the missing region from "a person riding a horse on the beach" to "a boy walking on the beach". HVITA (Qiu et al., 2020) is the only work that tackles such substantial context change, which occurs from the complete removal of a visual instance from the scene. However, HVITA has three major limitations: (1) HVITA mainly targets rectangle masks and thus lacks generalization to other mask forms, (2) the completed image produced by HVITA exhibits abrupt changes along the boundaries between the generated and original regions, and (3) HVITA has a heavy reliance on a refinement network to produce realistic images. To alleviate these issues, we propose a new framework called *Refill* that leverages a predicted semantic segmentation mask as guidance for image completion.

*Refill* performs image completion in three steps: 1) predicting the class of the missing instance, 2) generating a semantic segmentation mask of the missing region, and 3) completing the masked image using the segmentation guidance. Specifically, *Refill* predicts the class of the missing instance based on the context of the image, which is determined by mining the inter-instance co-occurrence using a transformer network. Then, *Refill* generates the segmentation masks of both the missing instance and the background area of the missing region individually using a conditional GAN and transformer body reconstruction network. Finally, by taking the generated segmentation mask as input, our framework generates a context-friendly instance and its background, which fills in the masked image to finally produce a realistic natural image. The proposed context-aware, segmentation-guided image completion framework enables *Refill* to handle missing regions with arbitrary shapes (such

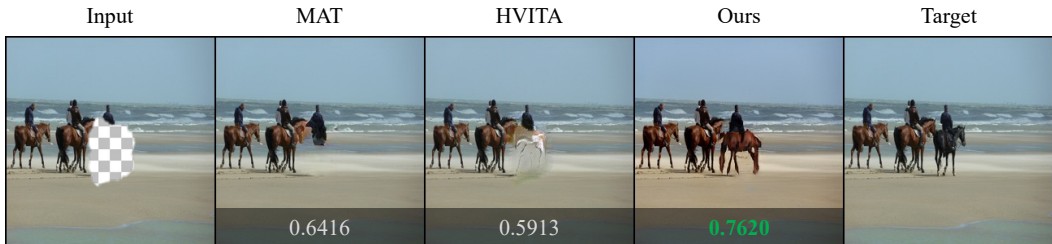

Context Query: *"A person riding a horse on the beach"*

Figure 1: From the first column, Input image with a missing region, results of state-of-the-art image completion approaches, such as MAT (Li et al., 2022) and HVITA (Qiu et al., 2020), Our result (*Refill*), and the target image. We compute CLIPScore around the generated part using the query text. As our approach generates a horse to complete the image rather than fills using background textures, CLIPScore of our result exhibits the best performance among the other models.

as scribbles), unlike HVITA (Qiu et al., 2020) which is best suited for missing rectangular regions. Note that *Refill* avoids the need for a refinement network, unlike HVITA, which heavily relies on the performance of its refinement network.

To evaluate and compare our model against existing methods, we first employ an off-the-shelf image captioning network, OFA (Wang et al., 2022), to produce a caption for each missing region of the masked images. We propose to use the produced caption as the *context query*, which represents the context of the missing regions. To measure how much the context of the image changes after completion, we propose to use two evaluation metrics: (1) CLIPScore (Schönfeld et al., 2021), which employs CLIP visual and textual encoders to determine whether the generated image region is well aligned with the context query, and (2) Visual Grounding Accuracy (VGA), which uses a pretrained visual grounding model (Wang et al., 2022) to determine whether the context query can successfully ground the generated image region. We also evaluate our method using conventional image quality assessment metrics, including FID (Heusel et al., 2017) and LPIPS (Zhang et al., 2018). On COCO-panoptic (Lin et al., 2014)/Visual Genome (Krishna et al., 2017) datasets, *Refill* shows comparable visual quality (FID=7.284/5.849) to the state-of-the-art image completion approach such as MAT (Li et al., 2022), while Visual Grounding Accuracy and CLIPScore are 12.472/14.107% and 0.027/0.029 better than HVITA. These results demonstrate that our approach can complete the missing regions of masked images in a context-friendly manner to yield high-quality images.

Our contributions are summarized as follows:

- We propose a novel framework called *Refill* which completes the missing region of masked images in a context-friendly manner, preserving the original context by leveraging a segmentation mask to encourage visual consistency between the generated and unmasked areas without relying on a refinement network.

- We present a novel combination of two transformer-based modules which facilitates our context-aware image completion pipeline. The missing instance inference transformer predicts the class of the missing instance effectively. The transformer-body background segmentation completion network shows better-recovered segmentation masks, especially under the presence of large missing regions.

- We propose to adopt CLIPScore and VGA to evaluate the context consistency between the original image and the completed image.

- *Refill* produces new visual instances in missing regions that are visually consistent with the unmasked areas. *Refill* also shows better performance in CLIPScore and VGA metrics compared to the baselines and exhibits comparable FID performance compared to the state-of-the-art approaches.

## 2    RELATED WORK

Traditionally, image completion task has been solved using approaches based on diffusion-based methods (Bertalmio et al., 2000; Ballester et al., 2001) and patch-based methods (Criminisi et al., 2003; 2004; Ding et al., 2018; Hays & Efros, 2007; Le Meur et al., 2011; Lee et al., 2016; Sun et al., 2005). The fundamental assumption of these models is that the missing parts of an image can be found and replaced with the remaining regions, making the model complete the missing region with only low-level features and repetitive patterns in the image.

With the advancement of deep learning, image completion models based on deep generative models have become the mainstream of photo-realistic image completion. Context encoder (Pathak et al., 2016) benefits from adversarial training inspired by Generative Adversarial Network (GAN) (Goodfellow et al., 2014) and shows perceptually more plausible results. VQGAN (Esser et al., 2021) and Palette (Saharia et al., 2022) introduce more advanced generative models: auto-regressive and denoising diffusion probabilistic modeling, and the approaches show compelling outputs. Along with efforts to introduce better generative models, there have been studies to improve the completion performance by revising existing architectures and convolution operations. Song et al. (2018a); Yan et al. (2018); Yu et al. (2018); Liu et al. (2019) propose contextual attention layers for long-range contextual encoding and image completion. Liu et al. (2018) propose a partial convolution layer to alleviate color discrepancy and blurriness in completed images. Yu et al. (2019) generalizes the partial convolution by introducing a dynamic feature selection mechanism at each spatial coordinate. Recent works tackle large-scale missing regions in masked images, a more challenging setting. Zhao et al. (2021) propose a modulation technique to handle this setting. Li et al. (2022) devises new transformer blocks to take advantage of long-range context interaction to complete images.

Generating photo-realistic images from semantic segmentation masks is also relevant work for image completion. The main goal is to condition the segmentation mask while preserving its spatial semantic information. To solve this issue, the authors of SPADE (Park et al., 2019) propose a spatially-adaptive (de)normalization layer by modulating pixel-wise features. Moreover, OASIS (Schönfeld et al., 2021) strengthens the discriminator by replacing conventional real or fake discriminator with a segmentation-based architecture. This mechanism improves the generator to synthesize realistic images well aligned with semantic layout. These works are originally not intended to complete masked images but designed to generate photo-realistic images from the noise vector conditioned on semantic segmentation masks. We bring SPADE and OASIS and give segmentation mask guidance for the missing region completion.

Other works employ semantic/structural information rather than directly complete the corrupted images. EdgeConnect (Nazeri et al., 2019) and E-CE (Liao et al., 2018) propose models that recover edge details first then fill out color next. Beyond using structural information, SPG-Net (Song et al., 2018b) and SG-Net (Liao et al., 2020) use semantic segmentation maps to contain more informative guidance than just edge. Our approach is close to SPG-Net and SG-Net, in that we recover the segmentation masks and generate the content. However, our method differs in that we handle more challenging scenarios where the instances are *entirely removed*. To our best knowledge, the only work for the context-aware image completion is HVITA (Qiu et al., 2020), where a target instance is wholly removed from an image. HVITA consists of four steps: (1) detecting visible instances, (2) constructing a graph using detected instances to understand the scene context, (3) generating a missing instance and placing it on the missing region, and (4) refining the inserted image. Yet, HVITA is the best suited for rectangular mask and show poor image quality, especially around the boundaries. In contrast, we are free to handle masks with arbitrary shapes since we directly generate new instances within missing regions guided by a completed segmentation map, unlike HVITA. The completed segmentation map helps to understand the context of images and encourages visual continuity on the boundaries between generated and unmasked regions. It also avoids relying on the refinement networks that is used in HVITA.

## 3    METHOD

Given an uncorrupted image $I$, we create a corrupted image $I_M$ using a binary mask $M$. The corrupted image can be expressed as $I_M = I \odot M$, where $\odot$ is element-wise multiplication. Our framework aims to complete the corrupted image $I_M$, where the visual instance is completely

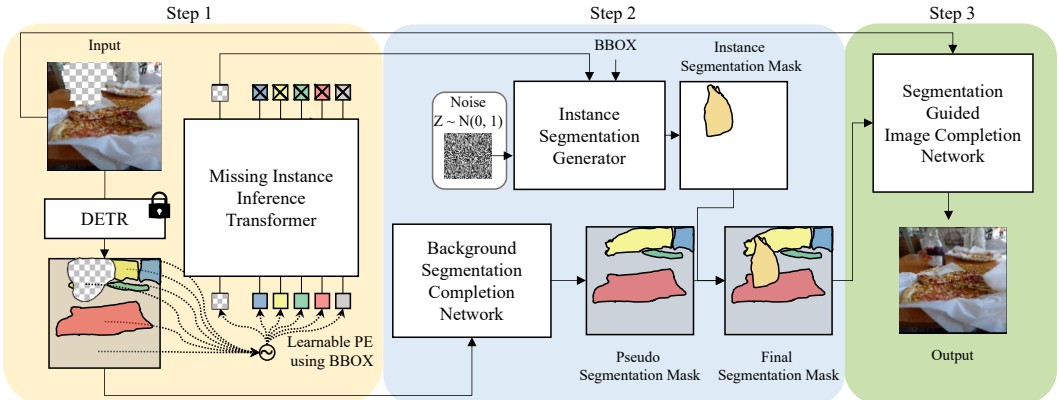

Figure 2: Overview of the proposed approach, called *Refill*. *Refill* completes the image in three steps: (1) infer the missing instance class, (2) complete a segmentation map in the missing region, and (3) translate the segmentation map to an image to hallucinate the missing region.

removed. This problem is challenging since the model should generate a target instance. At the same time, the generated instance should keep visual continuity with the existing parts.

Our framework (*Refill*) completes masked images in three steps. First, we infer a contextually appropriate instance by figuring out the category of the missing instance (Sec. 3.1). Second, we complete the missing semantic segmentation map based on the inference result from the previous step (Sec. 3.2). Finally, we transform the masked image with semantic segmentation into a realistic color image (Sec. 3.3). The framework enables each module to concentrate on the individually assigned task (instance inference, missing segmentation map completion, and image generation) instead of doing these in one shot. This is the key difference from most existing image completion approaches that mainly focus on the realism of completion results but do not consider semantics or which instance should be painted there. The overview figure of our framework is shown in Figure 2.

## 3.1 MISSING INSTANCE PREDICTION

To obtain the relationship information between instances in a given scene, *Refill* first predicts instance bounding box coordinates, instance classes, and a semantic segmentation map using the pre-trained DETR (Carion et al., 2020). Let the panoptic segmentation map $S_M = \text{DETR}(I_M)$, then we can extract box coordinates of the visible instances $B = [b_1, ..., b_k]$ and object classes $c = [c_1, ..., c_k]^\top$ from $S_M$, where $k$ is the number of predicted instances. Then, to infer the class of the missing instance $y_{target}$, *Refill* inputs the predicted classes of the visible instances along with a missing region token into a transformer network, called missing instance inference transformer. We first convert the visible instances' classes into learnable input tokens using a single linear layer. A quick approach is to utilize object queries from DETR as input tokens directly, but we observe such a method showed a worse performance compared to employing new learnable class embeddings. Furthermore, to inject the location information of the visible instances, we convert their bounding box coordinates into positional encoding vectors and sum them to the learnable class embeddings. To acquire the positional encoding vectors, we input the normalized center coordinate ($C_x$, $C_y$), width ($W$), and height ($H$) of the bounding box to a single linear layer where the activation is a sigmoid function. We also apply the same procedure for the missing region token, creating an embedding for missing region estimation. In addition, we also explore different ways to develop the positional encoding vectors and show that the adopted positional embedding mechanism provides the best missing instance estimation performance in Sec. 4.5.1. Below formulations are mathematical expressions of how our missing instance infer transformer works.

$$z_0 = \mathbf{E}_{class} + \mathbf{E}_{pos} = \text{MLP}(c') + \sigma(\text{MLP}(B')), \qquad z_0 \in \mathbb{R}^{(k+1)\times d} \qquad (1)$$

$$z'_l = \text{MSA}(\text{LN}(z_{l-1})) + z_{l-1}, \qquad l = 1, ..., L \qquad (2)$$

$$z_l = \text{MLP}(\text{LN}(z'_l)) + z'_l, \qquad l = 1, ..., L \qquad (3)$$

$$y = \text{LN}(z_L^0), \qquad (4)$$

where MLP is a multi-layer perceptron, $\sigma$ is a sigmoid activation, $d$ is the dimension of embedding vectors, $\boldsymbol{B}' = [\boldsymbol{b}_0] \cup \boldsymbol{B}$ where $\boldsymbol{b}_0$ is the missing region bounding box coordinate, and $\boldsymbol{c}' = [c_0] \cup \boldsymbol{c}$ where $c_0$ is a null class for missing region classification. The missing instance inference transformer consists of 12 transformer encoder layers ($L = 12$) with 8 heads. The missing region token interacts with the visible region tokens via the self-attention mechanism in the network. Thus, the network can predict the plausible class of the missing instance based on the detected instances and their location information. More details are in Appendix A.3.1.

## 3.2 SEMANTIC SEGMENTATION MAP GENERATION

Utilizing the predicted class of the missing instance from the previous step, we aim to generate the semantic segmentation map of the missing region. We create the segmentation map of both the instance and the background area individually with separate modules (instance segmentation generator and background segmentation completion network) and obtain the final segmentation map by inserting the missing instance segmentation into the background segmentation as shown in Figure 2.

We perform the missing instance segmentation generation using two modules: generator and discriminator, for the instance segmentation. The segmentation completion net aims to create a plausible segmentation map corresponding to the predicted instance class. For the implementation, we use the architecture from BigGAN (Brock et al., 2019), one of the most successful conditional GANs, with slight modification. We input the predicted missing instance class from the previous step and the box coordinates of the missing region to the Conditional Batch Normalization (De Vries et al., 2017) module in the instance segmentation generator. We train the model by using spectral normalization (Miyato et al., 2018) and hinge loss (Lim & Ye, 2017).

Second, the background segmentation completion network produces the segmentation map of the remaining region without attempting to generate the missing instance. To do this, we randomly scribble the ground truth segmentation map and let the background segmentation completion network restore it using cross-entropy loss. We experimentally found that this procedure can reconstruct background segmentation maps successfully. This module consists of convolution heads and tails with a transformer body. We explore that the transformer body helps to reconstruct the missing background segmentation map, especially large hole setting (See Sec. 4.5.2). Finally, we obtain the overall segmentation map of the missing region by inserting the instance segmentation into the background segmentation. More details are in Appendix A.3.3.

## 3.3 SEGMENTATION-GUIDED IMAGE COMPLETION

Using the predicted missing region segmentation map as guidance, a UNet (Ronneberger et al., 2015)-like completion model reconstructs the missing region of the masked image. The model takes in the masked image as input and outputs the restored version where SPADE (Park et al., 2019)/OASIS (Schönfeld et al., 2021) blocks force the model to reflect the semantics of the segmentation map. The details about the hyperparameters, objective function, and the model architecture are available in Appendix A.3.3. Since *Refill* can complete a masked image using any segmentation map, we can reconstruct diverse target instances of the same class by feeding different segmentation maps into the completion model. Figure 4 shows an example.

## 4 EXPERIMENT

### 4.1 DATASETS

We use two datasets called COCO-panoptic (Lin et al., 2014) and Visual Genome (Lin et al., 2014), which HVITA (Qiu et al., 2020) adopted to evaluate performance.

**COCO-panoptic.** COCO-panoptic (Lin et al., 2014) dataset contains 118K images for the train set and 5K images for the validation set with 80 things classes and 91 stuff classes. The dataset consists of natural images with multiple instances and has been hardly studied than center-aligned datasets (e.g., FFHQ (Karras et al., 2019), CelebA-HQ (Liu et al., 2015), and ImageNet (Deng et al., 2009)) in terms of image completion. For the experiment, we create a missing region using

two procedures: (1) first selecting a missing rectangular region that contains an instance, then (2) randomly cropping the image to be from 10% to 50% of the whole image, including the missing rectangular region. We randomly picked 50 points around the bounding box and drew thick lines between those points to make irregular regions. We select 30 classes from the things classes that are frequently observed in the images for the demonstration.

**Visual Genome.** We adopt Visual Genome (Krishna et al., 2017) to check the generalization ability of the proposed approach (*Refill*). The dataset contains 110k images with fine-grained 34k object categories. Due to the category mismatch between COCO-panoptic and Visual Genome, we use the DETR-predicted results as an annotation. Then, we make missing regions using the detected object boxes with the same size limitation used for creating the COCO-panoptic dataset.

## 4.2 BASELINE

**HVITA** (Qiu et al., 2020) is closely related work considering our problem setting. HVITA hallucinates the visual instance in three steps. First, the detection module detects visible instances, and the graph module predicts the features of a missing instance. Second, a conditional GAN generates a square-shaped RGB visual instance by conditioning the missing instance features and inserting them into the missing region. Finally, the refinement network improves visual continuity between the inserted part and the existing part. However, HVITA is originally designed to handle a rectangular mask and still suffers from abrupt discontinuity on the boundaries due to the object insertion step, as shown in Figure 3. We could not find any publicly available code, so we implemented HVITA carefully by referring to the details in the paper (Qiu et al., 2020).

**MAT** (Li et al., 2022) is one of the cutting-edge image completion models built on GAN framework. It consists of newly designed transformer blocks with a style manipulation module. We use the official author's implementation for training and evaluation.

## 4.3 METRICS

**LPIPS and FID.** There are several metrics for image quality measurement. L1, MSE, and PSNR quantify a pixel-wise error, and SSIM calculates the similarity between a generated image and the original one using luminance, contrast, and structure information. These metrics, however, are known to be inconsistent with human perception (Zhang et al., 2018). In addition, since there are various possible answers to complete the missing region, L1, MSE, and PSNR may give a high value, although completion results are plausible. To overcome this issue, we use LPIPS (Zhang et al., 2018), which is well known to agree with human perception by computing L2 distance on learned feature space (ImageNet-trained VGG). Moreover, we utilize Fréchet Inception Distance (FID) (Heusel et al., 2017) to assess the realism of completed images.

**CLIPScore.** To assess whether the hallucinated instance is perceptually well-aligned with the given text description, we employ CLIPScore computed using pretrained CLIP ViT-B/32 (Radford et al., 2021) visual and textual encoders. CLIPScore computes the cosine similarity between hallucinated instance embedding and the query text embedding. For the given hallucinated instance embedding $h$ and text description embedding $t$, we can compute $\text{CLIPScore}(h, t) = 2.5 \max(\cos(h, t), 0)$. This formulation is originally devised to evaluate image captioning performance in a reference-free manner (Hessel et al., 2021), but we use it to assess the perceptual alignment of hallucinated instances or entire generated images.

In contrast to LPIPS, which trims feature space using a uni-modal dataset with cross-entropy loss, CLIP organizes its feature space by optimizing Info-NCE loss (Sohn, 2016) on a cross-modal dataset made up of 400M (image and caption) pairs. Specifically, while the VGG backbone used for computing LPIPS gets supervision from pre-defined ImageNet classes Deng et al. (2009), CLIP encoders get strong feedback from vast amounts of natural images and texts on the internet. As the CLIP encoders show the capability of producing general representations via their superior zero-shot transfer ability (Radford et al., 2021), they can extract distinguishable features even in unseen classes.

**Visual Grounding Accuracy (VGA).** Locating the bounding box for a given query sentence in the image is the goal of visual grounding. If the model hallucinates a plausible instance similar to the original one, visual grounding model can place a bounding box around the hallucinated image given the text. For the evaluation, we adopt $\text{OFA}_{grounding}$ (Wang et al., 2022) trained on Visual

Table 1: Comparison of image synthesis quality on COCO-panoptic and Visual Genome datasets. All models are trained/finetuned on COCO-panoptic and evaluated on COCO-panoptic and Visual Genome (Zero-shot) images. We compare the quality of generated images using 4 metrics: CLIPScore, VGA, LPIPS and FID. $MAT_{pre}$ is pretrained on Places365-Standard (Zhou et al., 2017) that contains 8M images and is finetuned with our modified COCO dataset. $MAT_{scratch}$ and the other models are trained from scratch only using the modified COCO. We mark the best, the second-best in `normal yellow` and `light yellow` respectively.

| Metric | COCO-panoptic | | | | Visual Genome (Zero-shot) | | | |
|---|---|---|---|---|---|---|---|---|
| | CLIPscore ↑ | VGA (%) ↑ | LPIPS ↓ | FID ↓ | CLIPscore ↑ | VGA (%) ↑ | LPIPS ↓ | FID ↓ |
| $MAT_{pre}$ (Li et al., 2022) | 0.614 | 3.320 | 0.087 | 7.192 | 0.615 | 3.237 | 0.087 | 6.488 |
| $MAT_{scratch}$ (Li et al., 2022) | 0.606 | 3.216 | 0.093 | 7.895 | 0.608 | 3.141 | 0.093 | 7.302 |
| HVITA (Qiu et al., 2020) | 0.567 | 17.873 | 0.132 | 10.496 | 0.568 | 18.606 | 0.129 | 9.311 |
| $Refill_{spade}$ | 0.626 | 26.107 | 0.122 | 8.519 | 0.628 | 28.181 | 0.120 | 6.658 |
| $Refill_{oasis}$ | 0.641 | 30.345 | 0.119 | 7.284 | 0.644 | 32.713 | 0.116 | 5.849 |

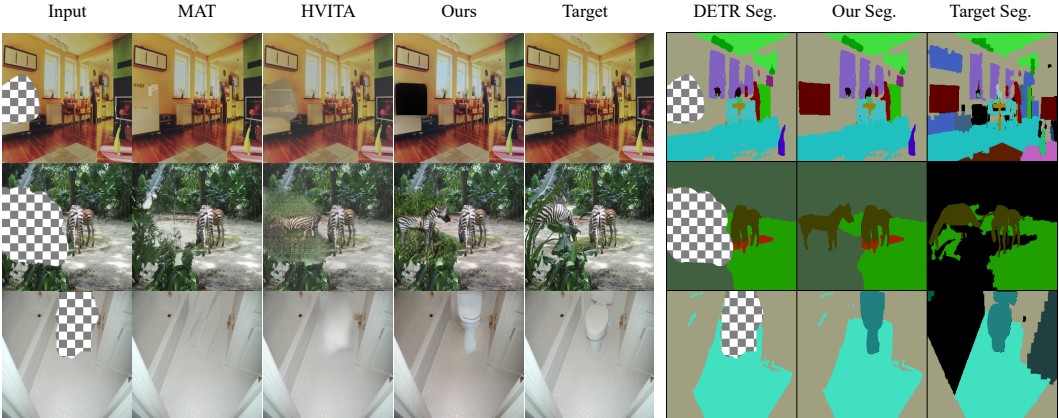

Figure 3: From 1st column to 5th column: Input, MAT (Li et al., 2022), HVITA (Qiu et al., 2020), Ours, GT. From 6th column to last column: DETR (Carion et al., 2020) seg, Our Seg and GT Seg. *Refill* restore the segmentation as shown in 6th column. We leave out more results on the Appendix A.4.

Genome Captions (Krishna et al., 2017), RefCOCO (Mao et al., 2016), and variants of RefCOCO, such as RefCOCO+ and RefCOCOg. The purpose of using VGA is similar to CLIPScore, but the core difference is the way how the feature space is annealed. The visual grounding model is trained using the 0.1M scale datasets, which is much smaller than the dataset used for CLIP, but the dataset contains high-quality and dense annotations.

Due to the lack of ground truth text descriptions for CLIPScore and VGA, we generate plausible sentences using a pretrained grounded image captioning network from $OFA_{captioning}$ (Wang et al., 2022). We measure VGA by setting mIoU threshold as 0.5.

## 4.4 EVALUATION RESULTS

To show the strengths of *Refill*, we perform the image completion task on the modified COCO-panoptic dataset (Sec. 4.1). We train $MAT_{pre}$ (Li et al., 2022), $MAT_{scratch}$ (Li et al., 2022), HVITA (Qiu et al., 2020), and our model using the dataset and evaluate the models using a test-split of COCO-panoptic dataset. The results are summarized in Table 1. $MAT_{pre}$ shows the lowest LPIPS and FID scores of 0.087 and 7.192, respectively. But it exhibits inferior results in CLIPscore and VGA on which *Refill* gives the best results (CLIPscore is 0.641, and VGA is 30.345%). In the case of HVITA, the most relevant baseline with our model, it can generate the target instance better than $MAT_{pre}$ and $MAT_{scratch}$. Still, the generated instances are less realistic than the other models, as seen in Table 1 and Figure 3. *Refill* is the model that compensates for the shortcomings of HVITA but reinforces the strengths. *Refill* attains comparable FID of 7.284 with the number of $MAT_{pre}$(7.192). At the same

| Input | Ours1 | Ours2 | Ours3 | Target |
|-------|-------|-------|-------|--------|

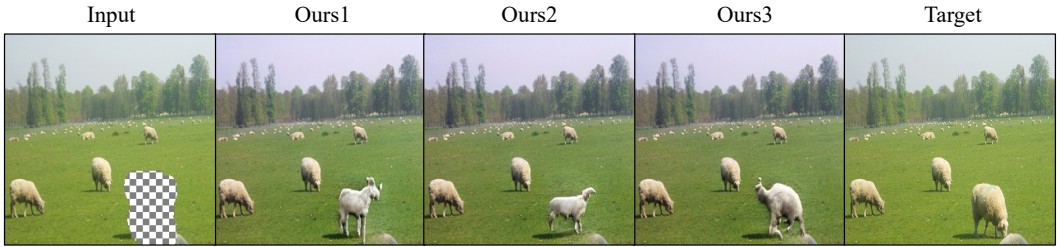

Figure 4: Example of diverse completion results of the proposed approach.

| Input | Ours | Target | | Input | Ours | Target |
|-------|------|--------|--|-------|------|--------|

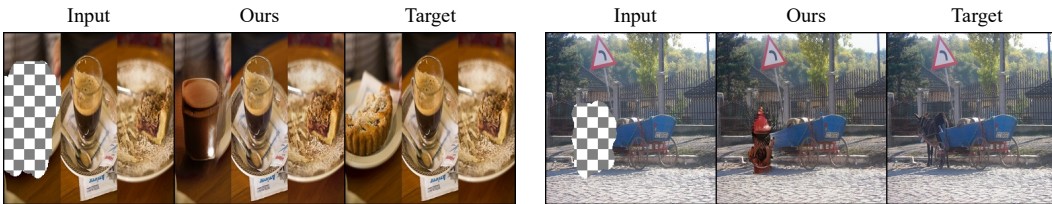

Figure 5: Failure cases. The proposed approach may fail to predict the same class of the target image, which may harm the evaluation scores. However, the generated images still follow the scene context. There are more failure cases in the Appendix A.4

time, *Refill* yields 0.027 and 12.472% better CLIPscore and VGA than the second best models. Note that $MAT_{pre}$ is trained on Places365-Standard, which is 80 times larger than the COCO-panoptic. The scale of the training data causes unfair evaluation between our model and $MAT_{pre}$, and the comparison with $MAT_{scratch}$ indicates better performance of *Refill* among the other completion models adopted in our experiments.

To test the generalization ability of each model, we evaluate the COCO-trained completion models on Visual Genome in the zero-shot fashion. As can be seen in Table 1, *Refill* proves the excellence in FID, CLIPscore, and VGA metrics except for LPIPS. Note that high LPIPS does not necessarily indicate low-fidelity results, where the results may include instances that differ from those of the original images as shown in Figure 4 and 5. Since LPIPS uses pixel-wise evaluation in the feature space, the model must generate an inpainting similar to the original image to obtain a low LPIPS score. However, our model completes the missing regions with visual instances of various shapes and classes, which are likely to differ from the original instance. Such diverse outputs result in comparatively higher LPIPS scores.

## 4.5 ABLATION STUDY

### 4.5.1 POSITIONAL ENCODING FOR MISSING-INSTANCE PREDICTION

Since instances' location information is as important as the class of instances to understand the scene, we devise six positional encoding variants. Let's denote relative bounding box coordinate as follow: $R_x = C_x - M_x$ and $R_y = C_y - M_y$, where $(M_x, M_y)$ and $(C_x, C_y)$ are missing region center coordinate and detected instances' bounding box center coordinate, respectively. $H$ and $W$ indicate the width and height of the bounding box. All the variables mentioned above are normalized to [0, 1]. ABS4C represents using $C_x, C_y, H, W$. ABS2C only uses $C_x$ and $C_y$. REL4C represents using $R_x, R_y, H$ and $W$. REL2C only uses $R_x, R_y$. No PE represents not using any positional encoding methods. Learnable means using the learnable positional encoding method. GCN indicates graph classification module in HVITA (Qiu et al., 2020).

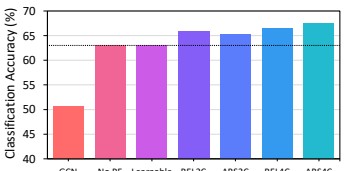

Figure 6: Comparison of the 6 positional encoding variants in missing instance infer transformer and GCN module in HVITA (Qiu et al., 2020) on COCO-panoptic

Among six methods, using the ABS4C method shows the best performance. Also, our missing instance inference transformer with ABS4C positional encoding shows 4.5% increase in classification accuracy performance compared to the No PE model. Moreover, using our transformer with ABS4C

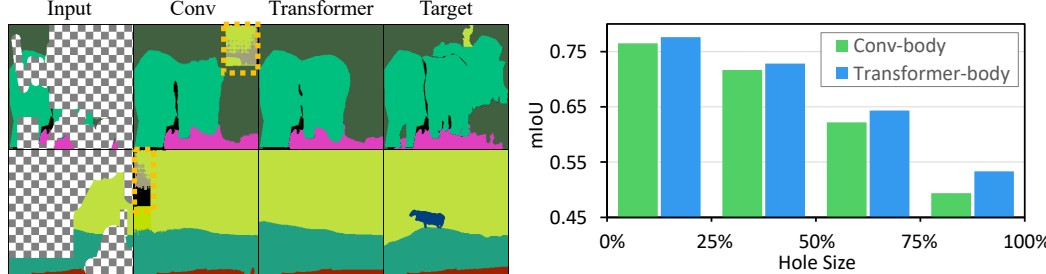

Figure 7: Predicting semantic masks given the severe missing regions. Transformer-based background completion network is more robust to the hole size than the Convolution-based version that often shows unwanted artifacts, as highlighted in yellow dotted boxes. The hole sizes of the first and the second row examples are 76.4% and 71.3%, respectively.

positional encoding significantly improves the performance of the GCN module in HVITA at 17%. See the details of six variants and their performance comparison in Figure 6.

### 4.5.2 BACKGROUND COMPLETION NETWORK

Inspired by MAT (Li et al., 2022), which uses a transformer architecture in the body of the network to complete large-scale masked images, we also adopt the transformer architecture in the body of the background segmentation completion network. Here, we use the Transformer body to complete missing background segmentation, not an image. As shown in Figure 7, transformer architecture helps reconstruct background semantic labels, especially for the large hole in the segmentation. As the hole size in segmentation from DETR increases, the mIoU performance gap between Transformer- and Conv-Based background segmentation completion network becomes more prominent. We also find that the Conv-based one generates artifacts on missing background segmentation in large hole settings, as shown in Figure 7. This demonstrates that Transformer architecture is better for recovering missing segmentation by leveraging global-range context interaction in a large hole setting. Both networks used the same number of parameters (44M) and were trained from scratch for this experiment.

### 4.6 FAILURE CASES

*Refill* may fail to generate correct class instances due to the wrong predicted class from the missing instance inference transformer. However, as shown in Figure 5, the generated instances are well harmonized with the remaining parts. In particular, *Refill* generates a cup and a fire hydrant instead of a cake and a horse. The images are still convincing.

## 5 CONCLUSION

This paper presents a novel framework called *Refill* for image completion, which hallucinates context-friendly visual instances instead of filling the missing region with surrounding textures. Through extensive experiments, we demonstrate the superiority of *Refill* in terms of CLIPScore and VGA metrics while achieving comparable FID scores over the baseline models. This work could be extended to more general scenarios, such as image completion for images with multiple missing regions or hallucinating multiple context-friendly instances within a single missing region.

**Limitation** The generation quality of our approach heavily depends on semantic image synthesis blocks (SPADE and OASIS), *Refill* inherits their shortcomings, especially generation diversity, as can be seen in Figure 4. Moreover, *Refill* is not aware of "*no instance*" and high-order geometry such as occlusion order and instance's orientation. Our approach is demonstrated with the 30 classes of COCO-panoptic things classes. Rather than using pre-defined categories, open-set segmentation and text-to-image translation would be intresting approaches. Solving these limitations could be interesting future works.

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

# A   APPENDIX

## A.1   ADDITIONAL EXPERIMENTS ON THE PROPERTIES OF MISSING INSTANCE INFERENCE TRANSFORMER

In this section, we perform diverse experiments to identify the effectiveness and the generalizability of the proposed missing instance prediction module. We first visualize the intermediate self-attention layers to verify the dependency between surrounding instances and the predicted instance. As shown in Figure 8, when our module predicts a missing class, the module gives more weight to the semantically related surroundings. We then perform another experiment to check whether our module's prediction also depends on the position of the missing region. Figure 9 describes how predicted classes change over different missing regions. From these two observations, we empirically conclude that our module rationally produces missing instance's class depending on the category and position of surrounding instances.

For the generalizability of our missing instance inference transformer, we adopt zero-shot setting and see if our module trained on COCO-panoptic dataset succeeds to produce accurate prediction class label of the unseen dataset, Visual Genome. Experiment results in Table 2 show that prediction accuracies in unseen dataset matches that of the seen dataset regardless of the choice in bounding box representation. Thus, we conclude our missing instance inference transformer well generalizes to unseen datasets as long as they share same class labels with the training set.

## A.2   ADDITIONAL EXPERIMENTS ON THE OBJECT REMOVAL TASK

While our work primarily focuses on generating a plausible instance in the missing region, the task of object removal, which focuses on overwriting the region with natural background, is also widely studied in the field of image inpainting. Here, we perform experiments to verify if our model is capable of object removal through small change in the pipeline. From the main framework in Figure 2, we omit pipeline for predicting and generating segmentation mask in Step 2 and use only background segmentation completion network. Experiment results from Table 3, suggest that object "*removed*" images look as realistic as instance "*inserted*" images which are generated according to original framework in this paper. Qualitative results can be found in Figure 10.

Table 2: Missing instance infer transformer performance on COCO-panoptic and VG (Zero-shot). The explanation of each methods including ABS4C, REL4C, ABS2C, REL2C, Learnable, No PE and GCN are described in Sec. 4.5.1.

|  | COCO-panoptic | Visual Genome (Zero-shot) |
|---|---|---|
| ABS4C | 67.548 | 67.236 (↓ 0.312) |
| REL4C | 66.466 | 66.283 (↓ 0.183) |
| ABS2C | 65.204 | 66.055 (↑ 1.149) |
| REL2C | 65.865 | 66.370 (↑ 0.505) |
| Learnable | 62.981 | 65.701 (↑ 2.720) |
| No PE | 63.041 | 65.047 (↑ 2.006) |
| GCN | 50.661 | 49.178 (↓ 1.483) |

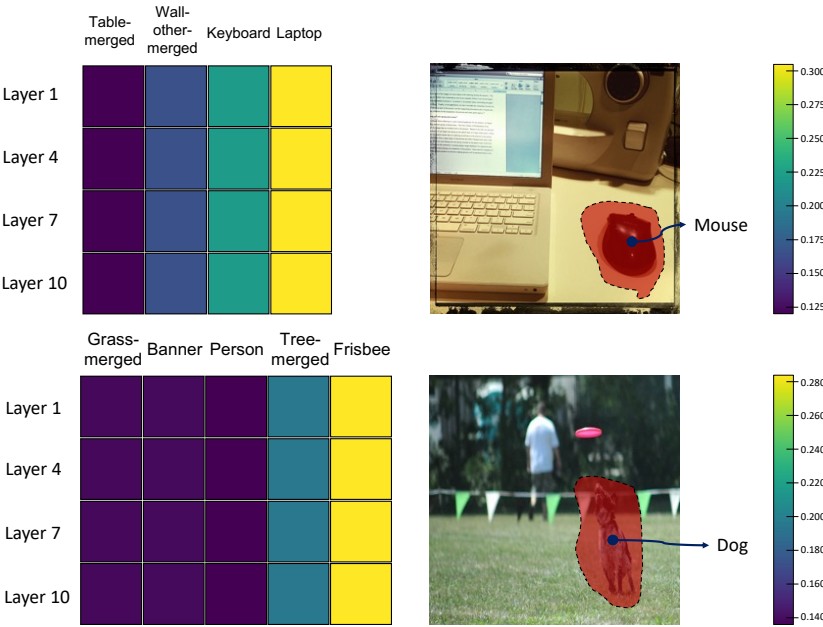

Figure 8: Self-attention visualization from the missing instance inference transformer. We visualize the self-attention of the missing instance token in the intermediate layer (Layer 1, Layer 4, Layer 7 and Layer 10) of the missing instance inference transformer. The missing region of each images is denoted by semi-transparent red mask.

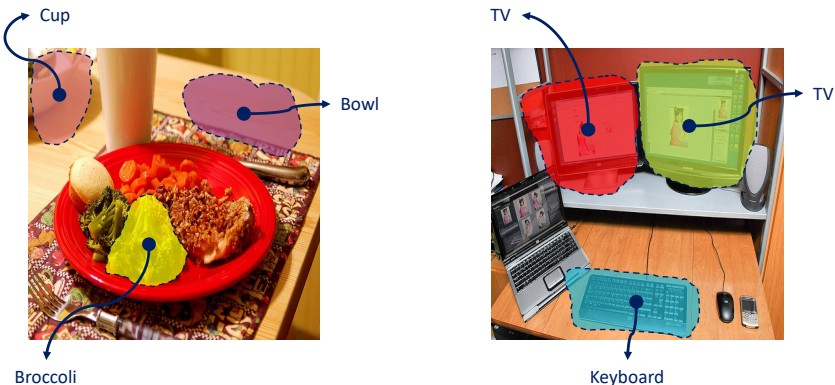

Figure 9: Missing instance class prediction changes depending on the position of each missing region. Each missing region in the images is denoted by different colors of masks. Predicted classes of each missing region are indicated with arrow.

Table 3: FID comparison between vanilla model in *Refill* and "*no instance*" version proposed in Appendix A.2. The vanilla model generates context-friendly visual instances in the missing region while the "*no instance*" version fills the missing region with surrounding textures by simply skipping the foreground instance prediction.

| Dataset | vanilla | "*no instance*" |
|---|---|---|
| COCO-panoptic | 7.284 | 7.874 (↑ 0.590) |
| Visual Genome (Zero-shot) | 5.849 | 6.887 (↑ 1.038) |

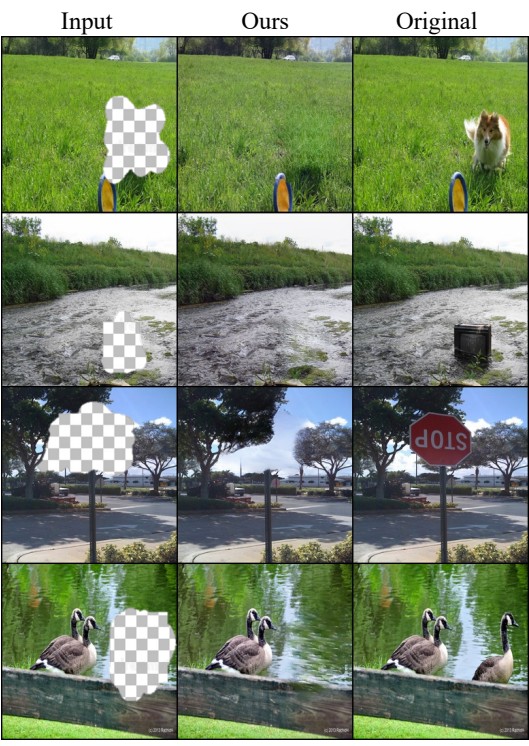

Figure 10: Qualitative results of *Refill*'s "*no instance*" version.

### A.3 NETWORK ARCHITECTURES AND TRAINING DETAILS

We implement our completion model for experiments using PyTorch (Paszke et al., 2019) library. We use PyTorch functions and define some necessary notations here to provide the architectural details of *Refill*.

EMBEDDING $(dim_{out})$ indicates PyTorch embedding function. TRANSENCLAYER $(dim_{token}, dim_{hidden}, head_{numb})$ is a vanilla transformer encoder layer. FC $(dim_{input}, dim_{output})$ is a single linear layer. CONV $(kernel_{size}, stride, padding)$ is a convolution layer. DBLOCK $(channel_{in}, channel_{out}, downsampling)$ and BIGGBLOCK $(channel_{in}, channel_{out}, upsampling, dim_{z_{split}}, dim_{shared})$ are BigGAN blocks implemented in StudioGAN libary (Kang et al., 2022). SELF-ATTENTION is a self-attention block implemented in StudioGAN. BN and LN indicate a batch normalization and layer normalization layer, respectively. SPADERESNETBLOCK is a SPADE block (Park et al., 2019). RESBLOCK-UP and -DOWN indicate ResNet blocks implemented in StudioGAN. UP$(size)$ indicates a upsampling function PyTorch provides. N is the number of classes in the segmentation mask. m and k indicate the batch size and the number of visible instances for a given masked image.

### A.3.1 MISSING INSTANCE INFERENCE TRANSFORMER

Missing instance inference transformer consists of 12 layers of transformer encoder layers (Vaswani et al., 2017) with 8 heads. The architectural details are described in Table 4. We use Adam optimizer (Kingma & Ba, 2015) with $\beta_1$ and $\beta_2$ of 0.9 and 0.999, respectively. The learning rates linearly increase until 50 epochs and linearly decrease until the end of the training (250 epochs).

### A.3.2 INSTANCE SEGMENTATION GENERATOR AND DISCRIMINATOR

We use BigGAN architecture (Brock et al., 2019) implemented in the StudioGAN library (Kang et al., 2022). The detailed BigGAN structure can be summarized in Tables 5 and 6, respectively. We use Adam optimizer with $\beta_1$ and $\beta_2$ of 0.0 and 0.999, respectively. The learning rates for the generator and discriminator are set to $5.0 \times 10^{-5}$ and $2.0 \times 10^{-4}$.

### A.3.3 BACKGROUND SEGMENTATION COMPLETION NETWORK

For the convolution-body background segmentation completion network, we use a UNet (Ronneberger et al., 2015) architecture. For the Transformer-body background segmentation completion network, we build 8 layers of the Transformer encoder and add 3 convolution layers and 3 ResNet blocks before and after the Transformer encoder layers. The details of the Transformer-body version are shown in Table 7. We optimize the whole segmentation completion network with Adam optimizer with a learning rate of $1.0 \times 10^{-4}$ and weight decay of $1.0 \times 10^{-4}$. $\beta_1$ and $\beta_2$ for Adam are set to 0.9 and 0.999, respectively.

### A.3.4 SEGMENTATION-GUIDED COMPLETION NETWORK

We use a UNet-like architecture for the segmentation-guided completion network. We apply SPADERESNETBLOCK to the decoder part to condition the completed segmentation mask into the UNet-like completion network. The architectural details of OASIS (Schönfeld et al., 2021) version of segmentation-guided completion network are explained in Tables 8 and 9. We follow the same training details as the original OASIS paper used with a single exception of the objective function. In order to preserve the unmasked region's data and generate contents only in the masked region, we apply the OASIS generator and discriminator loss only to the masked region and use additional L2 loss only on the unmasked region. We also add perceptual loss computed on the pre-trained VGG-19 recognition model to encourage fast convergence. For the SPADE (Park et al., 2019) version, we change the OASIS discriminator into the SPADE discriminator. Then, we follow the same training scheme described in the original SPADE paper.

Table 4: Architecture of missing instance inference transformer.

| Layer | Input | Output | Operation |
|---|---|---|---|
| Input Layer | (m, k+1, 1) | (m, k+1, 256) | EMBEDDING(1,256) |
| Hidden Layer | (m, k+1, 256) | (m, k+1, 256) | TRANSENCLAYER(256, 2048, 8) |
| Hidden Layer | (m, k+1, 256) | (m, k+1, 256) | TRANSENCLAYER(256, 2048, 8) |
| Hidden Layer | (m, k+1, 256) | (m, k+1, 256) | TRANSENCLAYER(256, 2048, 8) |
| Hidden Layer | (m, k+1, 256) | (m, k+1, 256) | TRANSENCLAYER(256, 2048, 8) |
| Hidden Layer | (m, k+1, 256) | (m, k+1, 256) | TRANSENCLAYER(256, 2048, 8) |
| Hidden Layer | (m, k+1, 256) | (m, k+1, 256) | TRANSENCLAYER(256, 2048, 8) |
| Hidden Layer | (m, k+1, 256) | (m, k+1, 256) | TRANSENCLAYER(256, 2048, 8) |
| Hidden Layer | (m, k+1, 256) | (m, k+1, 256) | TRANSENCLAYER(256, 2048, 8) |
| Hidden Layer | (m, k+1, 256) | (m, k+1, 256) | TRANSENCLAYER(256, 2048, 8) |
| Hidden Layer | (m, k+1, 256) | (m, k+1, 256) | TRANSENCLAYER(256, 2048, 8) |
| Hidden Layer | (m, k+1, 256) | (m, k+1, 256) | TRANSENCLAYER(256, 2048, 8) |
| Hidden Layer | (m, k+1, 256) | (m, k+1, 256) | TRANSENCLAYER(256, 2048, 8) |
| Hidden Layer | (m, k+1, 256) | (m, k+1, 256) | TRANSENCLAYER(256, 2048, 8) |
| Output Layer | (m, k+1, 256) | (m, k+1, 256) | FC(256, 256), GELU, LN(256) |

Table 5: Architecture for instance segmentation generator.

| Layer | Input | Output | Operation |
|---|---|---|---|
| Input Layer | (m,20) | (m,20480) | FC(20, 20480) |
| Reshape Layer | (m,20480) | (m,4,4,1280) | RESHAPE |
| Hidden Layer | (m,4, 4, 1280) | (m,8, 8, 640) | BIGGBLOCK(1280, 640, True, 20, 128) |
| Hidden Layer | (m,8, 8, 640) | (m,16, 16, 320) | BIGGBLOCK(640, 320, True, 20, 128) |
| Hidden Layer | (m,16, 16, 320) | (m,32, 32, 160) | BIGGBLOCK(320, 160, True, 20, 128) |
| Hidden Layer | (m,32, 32, 160) | (m,32, 32, 160) | SELF-ATTENTION |
| Hidden Layer | (m,32, 32, 160) | (m,64, 64, 80) | BIGGBLOCK(160, 80, True, 20, 128) |
| Hidden Layer | (m,64, 64, 80) | (m,64, 64, 1) | BN, RELU, CONV(80,3, 3, 1) |
| Output Layer | (m,64, 64, 1) | (m,64, 64, 1) | TANH |

Table 6: Architecture for instance segmentation discriminator.

| Layer | Input | Output | Operation |
|---|---|---|---|
| Input Layer | (m, 64, 64, 1) | (m, 32, 32, 80) | DBLOCK(3, 80, True) |
| Hidden Layer | (m, 32, 32, 80) | (m, 32, 32, 80) | SELF-ATTENTION |
| Hidden Layer | (m, 32, 32, 80) | (m, 16, 16, 160) | DBLOCK(80, 160, True) |
| Hidden Layer | (m, 16, 16, 160) | (m, 8, 8, 320) | DBLOCK(160, 320, True) |
| Hidden Layer | (m, 8, 8, 320) | (m, 4, 4, 640) | DBLOCK(320, 640, True) |
| Hidden Layer | (m, 4, 4, 640) | (m, 4, 4, 1280) | DBLOCK(640, 1280, False) |
| Hidden Layer | (m, 4, 4, 1280) | (m, 1280) | RELU, GSP |
| Output Layer | (m, 1280) | (m, 1) | FC(1280, 1) |

Table 7: Architecture for transformer-body background segmentation completion network.

| Layer | Input | Output | Operation |
|---|---|---|---|
| Input Layer | (m, 256, 256, 1) | (m, 256, 256, 64) | EMBEDDING |
| Hidden Layer | (m, 256, 256, 64) | (m, 128, 128, 128) | CONV(4, 2, 1) |
| Hidden Layer | (m, 128, 128, 128) | (m, 64, 64, 256) | CONV(4, 2, 1) |
| Hidden Layer | (m, 64, 64, 256) | (m, 32, 32, 512) | CONV(4, 2, 1) |
| Hidden Layer | (m, 32, 32, 512) | (m, 32, 32, 512) | TRANSENCLAYER(512, 2048, 8) |
| Hidden Layer | (m, 32, 32, 512) | (m, 32, 32, 512) | TRANSENCLAYER(512, 2048, 8) |
| Hidden Layer | (m, 32, 32, 512) | (m, 32, 32, 512) | TRANSENCLAYER(512, 2048, 8) |
| Hidden Layer | (m, 32, 32, 512) | (m, 32, 32, 512) | TRANSENCLAYER(512, 2048, 8) |
| Hidden Layer | (m, 32, 32, 512) | (m, 32, 32, 512) | TRANSENCLAYER(512, 2048, 8) |
| Hidden Layer | (m, 32, 32, 512) | (m, 32, 32, 512) | TRANSENCLAYER(512, 2048, 8) |
| Hidden Layer | (m, 32, 32, 512) | (m, 32, 32, 512) | TRANSENCLAYER(512, 2048, 8) |
| Hidden Layer | (m, 32, 32, 512) | (m, 32, 32, 512) | TRANSENCLAYER(512, 2048, 8) |
| Hidden Layer | (m, 32, 32, 512) | (m, 64, 64, 256) | UP(2), ResBlock(3, 1, 1) |
| Hidden Layer | (m, 64, 64, 256) | (m, 128, 128, 128) | UP(2), ResBlock(3, 1, 1) |
| Hidden Layer | (m, 128, 128, 128) | (m, 256, 256, 64) | UP(2), ResBlock(3, 1, 1) |
| Output Layer | (m, 256, 256, 64) | (m, 256, 256, 64) | RESHAPE, FC(64, 64), GELU, LN(64) |

Table 8: Architecture for segmentation-guided image completion generator (OASIS).

| Operation | Input | Size | Output | Size |
|---|---|---|---|---|
| CONCATENATE | $I_M$ | (m, 256, 256, 3) | $I_{M_{cat}}$ | (m, 256, 256, 4) |
| | $M$ | (m, 256, 256, 1) | | |
| CONV(4,1,1) | $I_{M_{cat}}$ | (m, 256, 256, 4) | $inter_0$ | (m, 256, 256, 32) |
| CONV(3,2,1) | $inter_0$ | (m, 256, 256, 32) | $inter_1$ | (m, 128, 128, 64) |
| CONV(3,2,1) | $inter_1$ | (m, 128, 128, 64) | $inter_2$ | (m, 64, 64, 128) |
| CONV(3,2,1) | $inter_2$ | (m, 64, 64, 128) | $inter_3$ | (m, 32, 32, 256) |
| CONV(3,2,1) | $inter_3$ | (m, 32, 32, 256) | $inter_4$ | (m, 16, 16, 512) |
| CONV(3,2,1) | $inter_4$ | (m, 16, 16, 512) | $inter_5$ | (m, 8, 8, 512) |
| CONV(3,2,1) | $inter_5$ | (m, 8, 8, 512) | $inter_6$ | (m, 4, 4, 512) |
| RESHAPE, FC(4*4*512,256) | $inter_6$ | (m, 4, 4, 512) | $mu$ | (m, 256) |
| RESHAPE, FC(4*4*512,256) | $inter_6$ | (m, 4, 4, 512) | $sigma$ | (m, 256) |
| NOISE SAMPLING, FC, RESHAPE | $mu$ | (m, 256) | $z$ | (m, 8, 8, 1024) |
| | $sigma$ | (m, 256) | | |
| CONCATENATE | $z_{3D}$ | (m, 64, 256, 256) | $z_y$ | (m, 64+N+1, 256, 256) |
| | $y$ | (m, N, 256, 256) | | |
| | $M$ | (m, 1, 256, 256) | | |
| SPADERESNETBLOCK, CONV(3, 1, 1) | $z$ | (m, 8, 8, 1024) | $up_0$ | (m, 8, 8, 512) |
| | $z_y$ | (m, 64+N+1, 256, 256) | | |
| CONCATENATE | $up_0$ | (m, 8, 8, 512) | $up_{0cat}$ | (m, 8, 8, 1024) |
| | $inter_5$ | (m, 8, 8, 512) | | |
| UP(2), SPADERESNETBLOCK | $up_{0cat}$ | (m, 8, 8, 1024) | $up_1$ | (m, 16, 16, 512) |
| | $z_y$ | (m, 64+N+1, 256, 256) | | |
| CONCATENATE | $up_1$ | (m, 16, 16, 512) | $up_{1cat}$ | (m, 16, 16, 1024) |
| | $inter_4$ | (m, 16, 16, 512) | | |
| UP(2), SPADERESNETBLOCK, CONV(3, 1, 1) | $up_{1cat}$ | (m, 16, 16, 1024) | $up_2$ | (m, 32, 32, 256) |
| | $z_y$ | (m, 64+N+1, 256, 256) | | |
| CONCATENATE | $up_2$ | (m, 32, 32, 256) | $up_{2cat}$ | (m, 32, 32, 512) |
| | $inter_3$ | (m, 32, 32, 256) | | |
| UP(2), SPADERESNETBLOCK, CONV(3, 1, 1) | $up_{2cat}$ | (m, 32, 32, 512) | $up_3$ | (m, 64, 64, 128) |
| | $z_y$ | (m, 64+N+1, 256, 256) | | |
| CONCATENATE | $up_3$ | (m, 64, 64, 128) | $up_{3cat}$ | (m, 64, 64, 256) |
| | $inter_2$ | (m, 64, 64, 128) | | |
| UP(2), SPADERESNETBLOCK, CONV(3, 1, 1) | $up_{3cat}$ | (m, 64, 64, 256) | $up_4$ | (m, 128, 128, 64) |
| | $z_y$ | (m, 64+N+1, 256, 256) | | |
| CONCATENATE | $up_4$ | (m, 128, 128, 64) | $up_{4cat}$ | (m, 128, 128, 128) |
| | $inter_1$ | (m, 128, 128, 64) | | |
| UP(2), SPADERESNETBLOCK, CONV(3, 1, 1) | $up_{4cat}$ | (m, 128, 128, 128) | $up_5$ | (m, 256, 256, 32) |
| | $z_y$ | (m, 64+N+1, 256, 256) | | |
| CONCATENATE | $up_5$ | (m, 256, 256, 32) | $up_{5cat}$ | (m, 256, 256, 64) |
| | $inter_0$ | (m, 256, 256, 32) | | |
| CONV(3,1,1), TANH | $up_{5cat}$ | (m, 256, 256, 64) | $I_F$ | (m, 256, 256, 3) |

Table 9: Architecture for segmentation guided image completion discriminator (OASIS).

| Operation | Input | Size | Output | Size |
|---|---|---|---|---|
| RESBLOCK-DOWN | $I$ | (m, 256, 256, 4) | $down_1$ | (m, 256, 256, 32) |
| RESBLOCK-DOWN | $down_1$ | (m, 128, 128, 128) | $down_2$ | (m, 64, 64, 128) |
| RESBLOCK-DOWN | $down_2$ | (m, 64, 64, 128) | $down_3$ | (m, 64, 64, 128) |
| RESBLOCK-DOWN | $down_3$ | (m, 32, 32, 256) | $down_4$ | (m, 32, 32, 256) |
| RESBLOCK-DOWN | $down_4$ | (m, 16, 16, 256) | $down_5$ | (m, 16, 16, 512) |
| RESBLOCK-DOWN | $down_5$ | (m, 8, 8, 512) | $down_6$ | (m, 4, 4, 512) |
| RESBLOCK-UP | $down_6$ | (m, 4, 4, 512) | $up_1$ | (m, 8, 8, 512) |
| CONCATENATE | $down_5$ | (m, 8, 8, 512) | $up_{1cat}$ | (m, 8, 8, 1024) |
|  | $up_1$ | (m, 8, 8, 512) |  |  |
| RESBLOCK-UP | $up_{1cat}$ | (m, 8, 8, 1024) | $up_2$ | (m, 16, 16, 256) |
| CONCATENATE | $down_4$ | (m, 16, 16, 256) | $up_{2cat}$ | (m, 16, 16, 512) |
|  | $up_2$ | (m, 16, 16, 256) |  |  |
| RESBLOCK-UP | $up_{2cat}$ | (m, 16, 16, 512) | $up_3$ | (m, 32, 32, 256) |
| CONCATENATE | $down_3$ | (m, 32, 32, 256) | $up_{3cat}$ | (m, 32, 32, 512) |
|  | $up_3$ | (m, 32, 32, 256) |  |  |
| RESBLOCK-UP | $up_{3cat}$ | (m, 32, 32, 512) | $up_4$ | (m, 64, 64, 128) |
| CONCATENATE | $down_2$ | (m, 64, 64, 128) | $up_{4cat}$ | (m, 64, 64, 256) |
|  | $up_4$ | (m, 64, 64, 128) |  |  |
| RESBLOCK-UP | $up_{4cat}$ | (m, 64, 64, 256) | $up_5$ | (m, 128, 128, 128) |
| CONCATENATE | $down_1$ | (m, 128, 128, 128) | $up_{5cat}$ | (m, 128, 128, 256) |
|  | $up_5$ | (m, 128, 128, 128) |  |  |
| RESBLOCK-UP | $up_{5cat}$ | (m, 128, 128, 256) | $up_6$ | (m, 256, 256, 64) |
| CONV(3,1,1) | $up_6$ | (m, 256, 256, 64) | $I_F$ | (m, 256, 256, N+1) |

## A.4 ADDITIONAL QUALITATIVE RESULTS

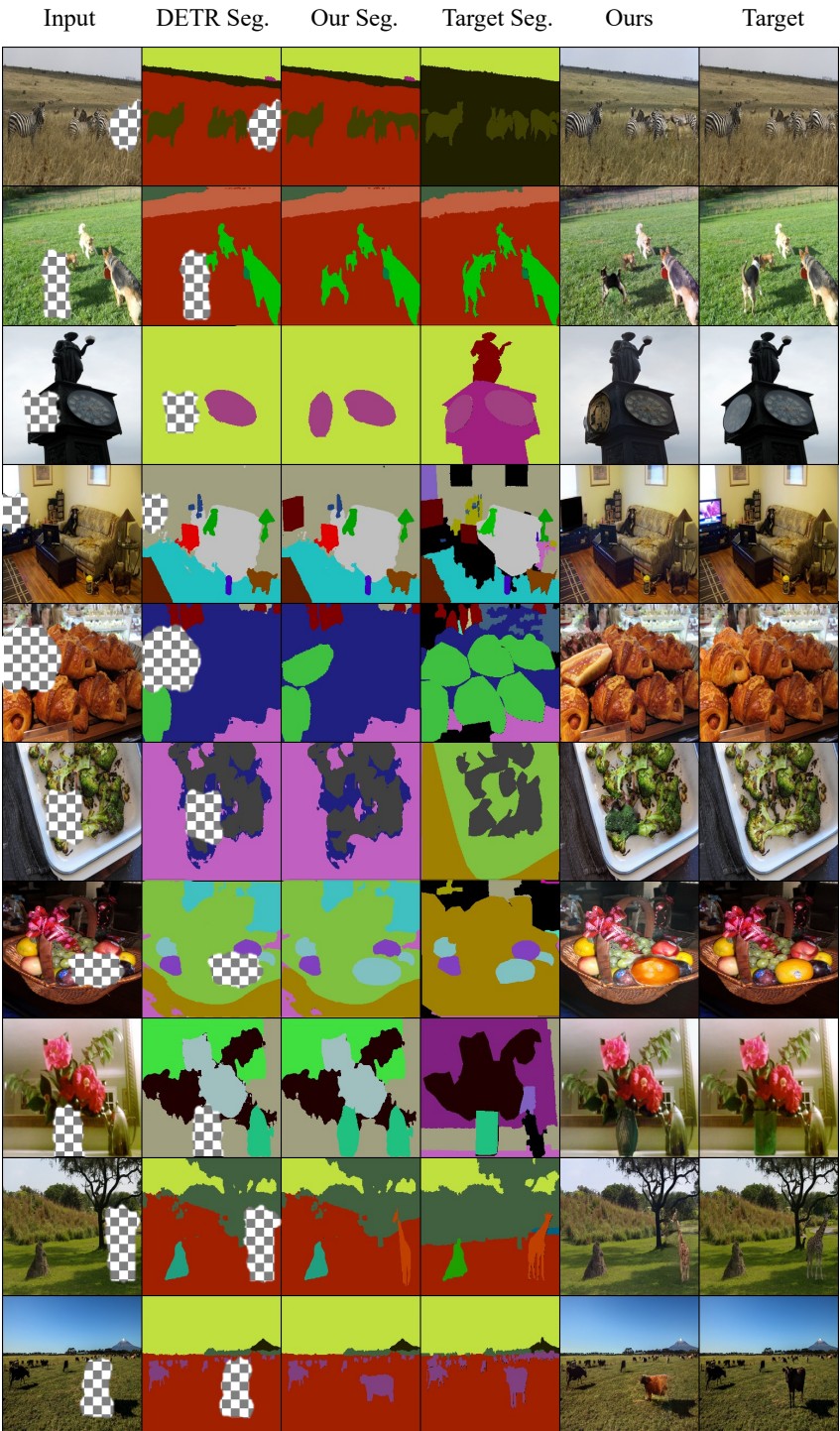

Figure 11: Image completion results.

Input DETR Seg. Our Seg. Target Seg. Ours Target

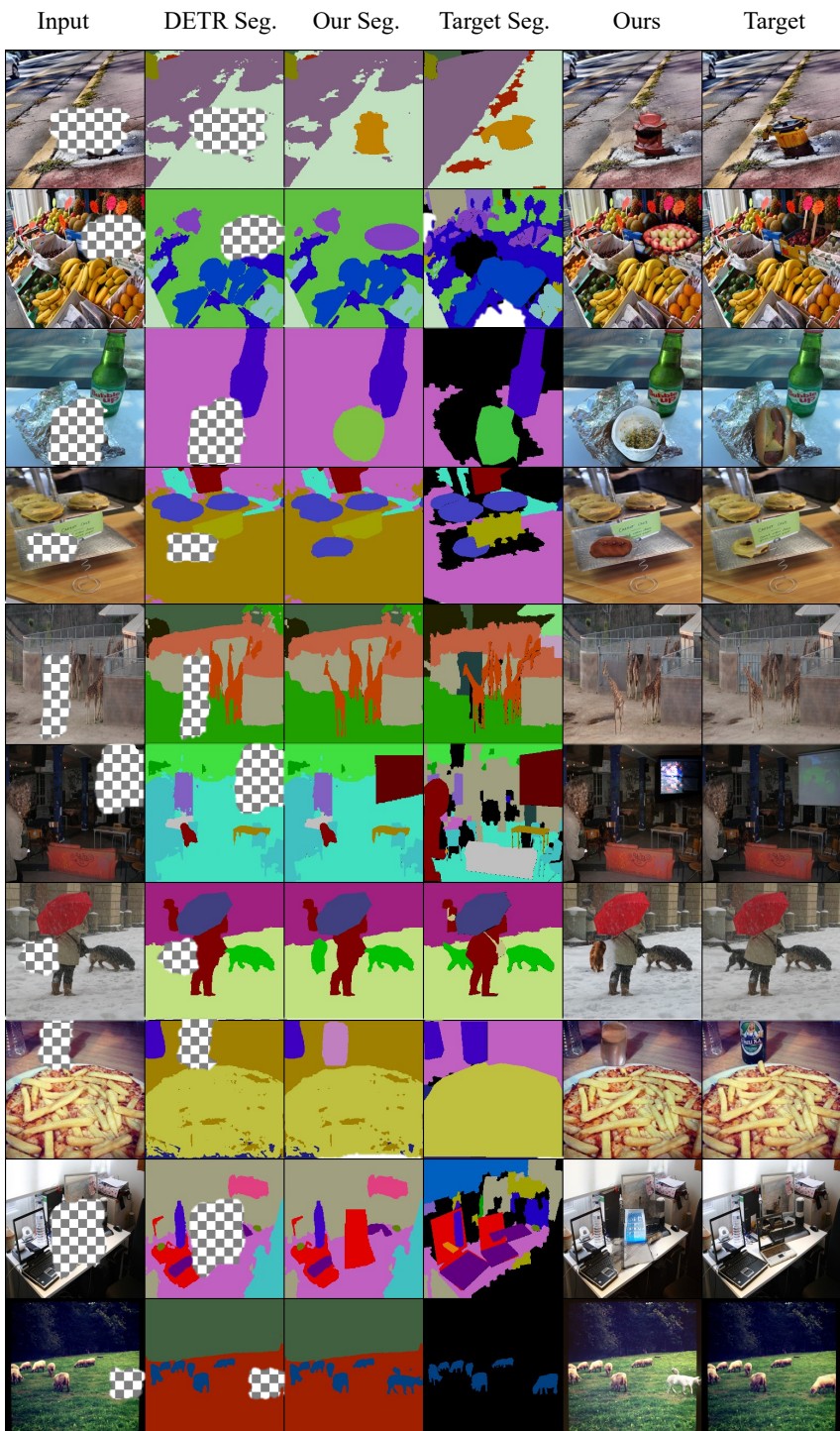

Figure 12: Image completion results.

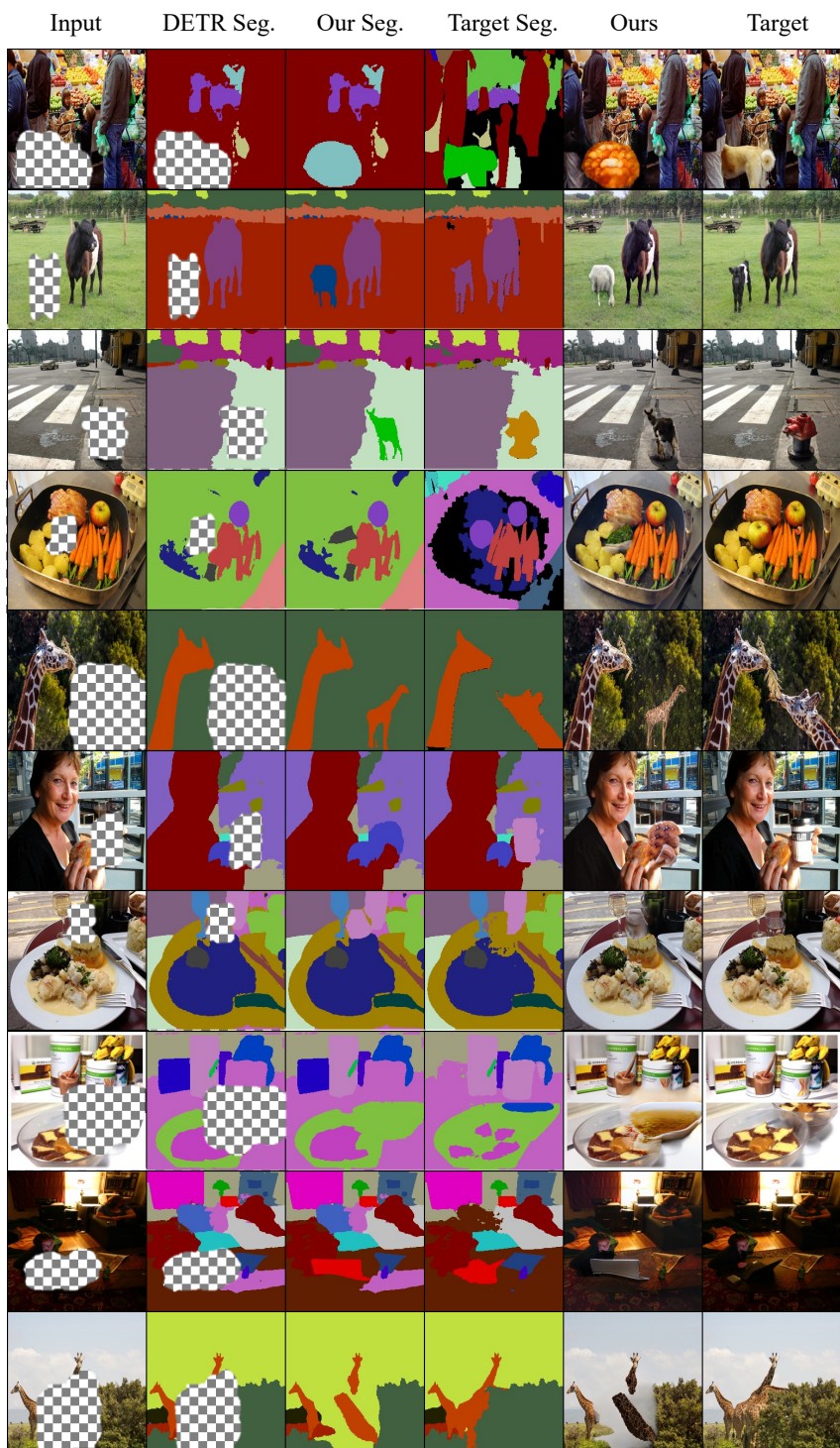

Figure 13: Failure case of *Refill*. From the first to the fourth row, *Refill* generates instances different from the target class. *Refill* does not aware the order of occlusions as shown in the fifth to the seventh row. Moreover, the eighth and ninth row shows our model suffers from reasoning the right scale and orientation. From the last row, the results show that segmentation mask is also crucial for the performance of generation ability.

