# OpenReview forum: "Context-Aware Image Completion"
_ICLR.cc/2023/Conference — Submitted to ICLR 2023_

### Official Review · Reviewer_HUfm · 2022-10-19

**Confidence:** 4
**Correctness:** 3
**Technical Novelty And Significance:** 2
**Empirical Novelty And Significance:** 2
**Recommendation:** 6

**Clarity, Quality, Novelty And Reproducibility:**

The paper is well organized and clearly written. The paper appears to be technically sound and maybe it is a little difficulty to reproduce the results.

**Strength And Weaknesses:**

Strength:
•	The novel image completion pipeline which tends to fill the missing region with new visual instances.
•	Previous context-aware image completion method only focuses on rectangular regions while this method are free to handle masks with arbitrary shapes.
•	Obtain high quality inpainting results and a number of quantitative results prove its effectiveness.

Weakness:
•	This method utilize a DETR pretrained on COCO which limits the application of this method. For some common used inpainting dataset like CelebA, FFHQ, Places, Paris street View, LSUN, it is hard to acquire the corresponding segmentation map to train the DETR.
•	The author claims ‘COCO-panoptic is more challenging than center-aligned datasets’, but the author should prove the generalization of the proposed method on other domain datasets like face (FFHQ)、buildings (Paris Street View).
•	One important application of inpainting is object removing while this method aims for new object generation. What about the performance of the proposed method on the object removing?
•	I find in most figures (Fig 1, Fig 3-5, Fig 8-9), there exist an obvious color discrepancy between the ground truth and the results of this paper. Can you make an explanation?
•	When processing large missing area, does the segmentation completion network still has the ability to generate new objects or just inpaint with background contents?
•	How is the model complexity and inference time compared to other methods?



**Summary Of The Paper:**

Different from the common inpainting methods, this paper propose a novel image inpainting model tending to fill the missing region with new visual instances instead of filling in the background content. The author first use two transformer-based networks to inpaint the background segmentation map and foreground segmentation map respectively. Then, an unet network is used to hallucinate the missing area with the help of the combined segmentation map. Besides, the author adopts the CLIPScore and VGA to evaluate the context consistency between the original image and the completed image.

**Summary Of The Review:**

In summary, this paper proposes a novel image completion method which shows advantage in hallucinating missing region with new visual instances. Substantial experiments prove the effectiveness of the proposed method.

---

> ### Author Response · Authors · 2022-11-25
> **Response to Reviewer HUfm (2/2)**
>
> > [“No Instance” Performance]
>
> > Q.3. One important application of inpainting is object removing while this method aims for new object generation. What about the performance of the proposed method on the object removing?
>
> Refill can perform “object removing”  by simply skipping the foreground object prediction.
> We found that the Refill’s object removal performance is visually convincing, as shown in **Appendix Figure 10**. In addition, we provide the FID score of object removal and further details in **Table 3 and section A.2 of the Appendix**, where the foreground object prediction is discarded, and the background prediction only is used. FID of Refill’s object removal is comparable to the other baselines(Vanilla Refill, MAT, and HVITA) even though we don't originally intend to design our pipeline for the purpose of object removal. The aforementioned qualitative and quantitative results demonstrate the effectiveness of Refill in removing objects.
>
> We have added **Appendix A.2 section** and added **Figure 10 and Table 3 in the Appendix** during the revision period.
>
> > [Color Discrepancy]
>
> > Q.4. I find in most figures (Fig 1, Fig 3-5, Fig 8-9), there exist an obvious color discrepancy between the ground truth and the results of this paper. Can you make an explanation?
>
> We acknowledge the color-shifting problem in unmasked regions after completion. We thought this is caused by pixel-wise modulation even in unmasked region from SPADE/OASIS blocks. In order to preserve the original contents of the unmasked regions and generate contents only in the masked region, we apply the OASIS generator and discriminator loss only to the masked region and use additional L2 loss only on the unmasked region. We also add perceptual loss computed on the pre-trained VGG-19 feature space to encourage fast convergence. Moreover, to learn specific modulation depending on the mask regions, we assigned an extra class label to the masked regions of the segmentation maps. As it takes longer training time, we are going to share the results as soon as possible. Thanks.
>
>
> > [Segmentation Completion Network Performance for large hole setting]
>
> > Q.5. When processing large missing area, does the segmentation completion network still has the ability to generate new objects or just inpaint with background contents?
>
> Yes, Refill’s segmentation completion network still has the ability to complete segmentation masks. Although the performance degrades as the hole size increases, the transformer layer helps to reconstruct the segmentation mask, especially in large hole settings, by leveraging global-range feature interaction (See **Figure 7** in the main paper).
>
> > [Reproducibility]
>
> We are planning to open the source code for this project once the paper is published.

---

> ### Author Response · Authors · 2022-11-25
> **Response to Reviewer HUfm (1/2)**
>
> We thank the reviewer HUfm for the thoughtful comments on our study. In this phase, we try to answer some concerns raised by the reviewer carefully.
>
> > [DETR dependency]
>
> > Q.1. This method utilize a DETR pretrained on COCO which limits the application of this method. For some common used inpainting dataset like CelebA, FFHQ, Places, Paris street View, LSUN, it is hard to acquire the corresponding segmentation map to train the DETR.
>
> Most conventional image inpainting works to address the restoring image when the parts of an instance are masked out. In particular, they have focused on center-aligned datasets (such as FFHQ and CelebA) in which parts of the face are missing. Also, they have shown  pluralistic generation in background-dominant scenes such as buildings and natural photography (Paris Street View and Places 2). In order to restore these images, instance-level information from a recognition model is not necessarily needed. Feature-level information from a stack of learning-based models is sufficient to restore the parts of the instance or background.
>
> In this paper, we cope with a hardly addressed image inpainting scenario, where the entire visual instance is removed. Only using feature-level information leads the model to lack the ability to generate scene/context-friendly instances in missing regions. To this end, we conclude to leverage instance-level context from DETR, similar to HVITA, which is the only prior work of this image inpainting scenario. Moreover, to learn ample amounts of inter-instance relations between multiple instances in the scene and ensure the removal/restoration of various types of missing instances, we use COCO-panoptic and Visual Genome datasets.
>
> Please note that the dataset (CelebA, FFHQ, Places, Paris Street View, LSUN) is not a proper dataset for this problem set since these datasets usually consist of few-instance/single-instance images. Hopefully, as Refill is able to reconstruct the segmentation map, Refill can be adopted on the aforementioned dataset by skipping the missing instance inference step of Refill if the GT segmentation is prepared.
>
> Moreover, we acknowledge that it would be a great future work direction to devise a new framework to generate plausible visual instances in the missing region without instance-level recognition results (segmentation).
>
> Please note that we change the title of the paper “Context-Aware Image Completion” to “Instance-Aware Image Completion” to emphasize the difference between conventional image inpainting tasks and our project.
>
> > [Overclaimed sentence ]
>
> > Q.2. The author claims ‘COCO-panoptic is more challenging than center-aligned datasets’, but the author should prove the generalization of the proposed method on other domain datasets like face (FFHQ)、buildings (Paris Street View) The author claims ‘COCO-panoptic is more challenging than center-aligned datasets’, but the author should prove the generalization of the proposed method on other domain datasets like face (FFHQ)、buildings (Paris Street View)
>
> We changed the sentence “The dataset consists of natural images with multiple instances, so COCO-panoptic is more challenging than center-aligned datasets.” into “The dataset consists of natural images with multiple instances and has been hardly studied than center-aligned datasets.” in **Sec.4.1** of the main paper. Moreover, FFHQ and Paris Street View are not well-suitable datasets considering our completion scenario as we consider the instance-level context. (See the reply of **[DETR dependency]** for further details.)

---

> ### Author Response · Authors · 2022-12-11
> **Hope to continue further discussions with Reviewer HUfm**
>
> Thank you again for your insightful remarks.
>
> Your concerns, notably those including "DETR dependency," "correcting overclaimed sentence," "reproducibility," "no instance performance," and "segmentation completion network performance for large hole setting," were carefully considered, and we made an effort to address them.
>
> We are happy to hear your comments for further discussion. Thanks.

---

### Official Review · Reviewer_xwWL · 2022-10-24

**Confidence:** 4
**Correctness:** 3
**Technical Novelty And Significance:** 2
**Empirical Novelty And Significance:** 3
**Recommendation:** 5

**Clarity, Quality, Novelty And Reproducibility:**

The writing is easy to follow and the task is interesting, but the technical contribution and experiments still need to improve.

**Strength And Weaknesses:**

Strengths
1)	To predict the missing semantic instance and hallucinate it is a challenging but interesting problem.
2)	The writing is easy to follow.
3)	The two introduced evaluation metrics are reasonable and prove the completed context's validity.

Weakness
1)	As I understand, the model performance is highly dependent on the missing instance prediction module. Still, it lacks sufficient explanation and analysis to prove the effectiveness, generalization, and robustness of this module. After all, the semantic instance inference in the segmentation map, especially the corrupted segmentation map is challenging. How about the performance of real-world images?
2)	From the results, the synthesized content is not consistent in style with the original context, and the completed method needs to be improved.
3)	Although the proposed method is superior in terms of the two new metrics, it is worse in the metrics of LPIPS and FID. The (novel) application of image completion, which originally intended to complete images to generate realistic content, should be clarified more clearer.


**Summary Of The Paper:**

This submission proposes an image completion method to restore the missing semantic instance while preserving the relationship with the original context. To this end, three steps are proposed to complete the inpainting process, including predicting the semantic instance, generating the instance mask, and completing the image. Two evaluation metrics are introduced to assess the completed context.

**Summary Of The Review:**

See above

---

> ### Author Response · Authors · 2022-11-25
> **Response to Reviewer xwWL (2/2)**
>
> > [Not consistent style]
>
> > Q.2. From the results, the synthesized content is not consistent in style with the original context, and the completed method needs to be improved.
>
> We acknowledge the color-shifting problem in unmasked regions after completion. We thought this is caused by pixel-wise modulation even in unmasked region from SPADE/OASIS blocks. In order to preserve the original contents of the unmasked regions and generate contents only in the masked region, we apply the OASIS generator and discriminator loss only to the masked region and use additional L2 loss only on the unmasked region. We also add perceptual loss computed on the pre-trained VGG-19 recognition model to encourage fast convergence. Moreover, to learn specific modulation depending on the mask regions, we assigned an extra class label to the masked regions of the segmentation maps.  As it takes longer training time, we are going to share the results as soon as possible. Thanks.
>
> > [Clarify worse LPIPS and FID]
>
> > Q.3. Although the proposed method is superior in terms of the two new metrics, it is worse in the metrics of LPIPS and FID. The (novel) application of image completion, which originally intended to complete images to generate realistic content, should be clarified more clearer.
>
> We point out that our model, Refill, is not worse in terms of FID but rather superior, as it produces the best FID value on Visual Genome in a zero-shot setting and on COCO-panoptic where the models are trained from scratch. Refill comes down to second place only when compared with MAT_pre, which is pretrained on a dataset **80 times larger** than COCO-panoptic (Places365-Standard), while Refill is trained on COCO-panoptic only. Furthermore, despite having been trained on a way smaller dataset, Refill shows a comparable FID number: 7.192 (Refill) vs. 7.284 (MAT_pre). Such results show Refill’s superiority in producing realistic images.
>
> We insist that a high LPIPS does not necessarily indicate low-fidelity results, especially in the case of Refill, where the results may include instances that differ from those of the original images, as shown in **Figures 4 and 5** of the main paper.
>
> Since LPIPS uses pixel-wise evaluation in the feature space, the model must generate an inpainting similar to the original image to obtain a low LPIPS score. However, our model completes the missing regions with visual instances of various shapes and classes, which are likely to differ from the original instance. Such diverse outputs result in comparatively higher LPIPS scores. Nevertheless, this does not necessarily mean that the generated instances are unrealistic but just different from the original instances. Furthermore, the low FID scores indicate that Refill produces high-fidelity images, and convincing visual instances are shown in **Figures 3, 4, 11, and 12**.
>
> We also refer to “Mask-Aware Transformer for Large Hole Image Inpainting, CVPR 2020,” which states the limitation of LPIPS in that “pixel-wise evaluation still greatly punishes diverse inpainting systems for large holes”. Thus, the LPIPS can penalize our model, which produces diversified inpaintings.
>
> We revised the main paper in **Sec 4.4** for the clarification of the high LPIPS of Reill.

---

> ### Author Response · Authors · 2022-11-25
> **Response to Reviewer xwWL (1/2)**
>
> We would like to thank reviewer xwWL’s valuable comments, especially for acknowledging the usefulness of the proposed evaluation metrics (CLIPScore and VGA). We try to answer the major concerns raised by the reviewer carefully.
>
> > [Additional exp for missing instance prediction module and segmentation completion module]
>
> > Q.1.  As I understand, the model performance is highly dependent on the missing instance prediction module. Still, it lacks sufficient explanation and analysis to prove the effectiveness, generalization, and robustness of this module. After all, the semantic instance inference in the segmentation map, especially the corrupted segmentation map is challenging. How about the performance of real-world images?
>
> We perform diverse experiments to identify the effectiveness and the generalization ability of the proposed missing instance prediction module. We first visualize the intermediate self-attention layers to investigate the relations between surrounding instances and the predicted instance. As can be seen in **Appendix Figure 8**, when our module predicts a missing class, the module gives more weight to the semantically related surroundings. We then perform another experiment to check whether our module’s prediction is also dependent on the position of the missing region. **Appendix Figure 9** describes how predicted classes change over different missing regions. From these two observations, we empirically conclude that our module can rationally hallucinate a missing instance’s class based on the category and relative position information of surrounding instances.
>
> For the generalization ability of our missing instance inference transformer, we adopt a zero-shot setting and see if our module **trained on the COCO-panoptic dataset succeeds in predicting the class label of a missing region in an unseen dataset, Visual Genome**.  Experiment results in **Appendix Table 2** indicate that prediction accuracy of ABS4C method still ranked the best in unseen dataset. Thus, we conclude our missing instance inference transformer generalizes well to unseen datasets as long as they share the same class labels with the training set.
>
> Moreover, in order to investigate the performance of our segmentation completion network, we conduct an apple-to-apple **comparison with segmentation output by just using DETR** based on the segmentation quality metric (mIoU). As shown in the Table below, Refill is capable of completing more precise segmentation masks than DETR on both COCO-panoptic (seen) and Visual Genome (unseen) datasets.
>
> ||COCO-panoptic|Visual Genome(Zero-shot)|
> |:---:|:---:|:---:|
> |DETR naive seg|0.5211|0.5821|
> |Refill Completed seg|0.5688|0.6362|
>
> We have added the **Appendix A.2, Figure 10 and Table 3** by summarizing above replies.

---

> ### Author Response · Authors · 2022-12-11
> **Hope to continue further discussions with Reviewer xwWL**
>
> Thank you again for your thoughtful comments.
>
> We tried to elucidate your concerns especially on the "need for additional experiments about missing instance inference transformer" and "clarification for low LPIPS and FID." For the further discussion, we are eager to get your responses. Thanks.

---

### Official Review · Reviewer_675z · 2022-11-03

**Confidence:** 4
**Correctness:** 3
**Technical Novelty And Significance:** 2
**Empirical Novelty And Significance:** 2
**Recommendation:** 5

**Clarity, Quality, Novelty And Reproducibility:**

Lack of academic novelty. In terms of weakness 3, the authors may argue that predicting objects is the major contribution of the paper. However, that also means predicting a limited set of objects in limited scenarios is the only thing that the proposed method can do. Indeed it is an improvement over HVITA, but it is more like an incremental one, that follows the path of HVITA and equipped with several off-the-shelf components.

**Strength And Weaknesses:**

Strength
1. The proposed pipeline is straightforward and effective.
2. Improve over HVITA by eliminating the restriction of rectangular missing region.


Weakness
1. Heavily limited to the pretrained detector. The proposed pipeline relies on DETR to provide the objects in the image, subsequent modules is limited to the pre-defined object categories of DETR. In the experiment design, only supported objects are masked as missing. Such heavy dependency limits the real-world applications of the method.
2. The missing region is within or around the instance bbox, and the method predicts one object regardless of the existence or the number of objects. In many inpainting scenario, the missing region doesn't does not necessarily contain an object in the middle of the region.
3. Experimental design is biased.
3.1 The datasets are prepared in favor of the proposed method, i.e., COCO and Visual Genome contains many objects; The missing regions are created around a carefully selected subset of objects.
3.2 The evaluation metric is biased. The CLIPScore and VGA depends on the object in the missing region. It is obvious that any inpainting method fails as long as it doesn't predict a correct category of object in the missing region.
3.3 It will be more convincing to provide methods to determine whether there are objects in the missing ("no instance"), and how to deal with cases that half of an object is missing while the other half is not ("instance across missing region").
3.4 Meanwhile, experiments should be conducted on the full (not selected by object category) COCO-panoptic and Visual Genome dataset, with random missing region instead of around object bbox.
4. It will be interesting to show whether the predicted instance in the impainted image can be detected with DETR. If yes, is the predicted category the same as the output of the "Missing Instance Inference Transformer"?

**Summary Of The Paper:**

The paper proposed a pipeline for image completion, which fills the missing region with a hallucinated instance. It consists of four parts. 1. Object detection via DETR within the masked image. 2. Predict the missing object's category via a multi-head attention network. 3. Generate the foreground & background segmentation mask with GAN. The final segmentation mask is foreground + background. 4. Segmentation guided image completion like SPADE / OASIS.

Experiments are performed on COCO-panoptic and Visual Genome dataset. Four metrics are used, including LIPIPS, FID, CLIPScore, Visual Grounding Accuracy (VGA).

**Summary Of The Review:**

I do not recommend acceptance of the paper because of lack of academic novelty. The paper provides an incremental improvement over HVITA, but does not solve critical problems like "no instance" or "instance across missing region".

---

> ### Author Response · Authors · 2022-11-25
> **Response to Reviewer 675z (3/3)**
>
> > [Biased Evaluation Metric]
>
> > Q.3.2. The evaluation metric is biased. The CLIPScore and VGA depends on the object in the missing region. It is obvious that any inpainting method fails as long as it doesn't predict a correct category of object in the missing region.
>
>
>  We acknowledge that CLIPScore and VGA favor Refill, which has been specifically trained to reconstruct target visual instances, over the other baselines. However, we politely remind the reviewer that the goal of this paper is completing images by generating context-friendly/scene-aware visual instances in the masked region rather than filling them out with the surrounding texture. Thus, our framework inevitably requires evaluation metrics that can detect whether the intended object is synthesized successfully or not.
>
>  Furthermore, we point out that we also measure FID and LPIPS (please refer to **Table 1** in the main paper), which are generally impartial to object existence. Since FID is widely used to quantify image realism, it can make up for CLIPScore and VGA’s shortcoming that “those metrics highly depend on the object in the missing region rather than image realism”
>
>  Refill exhibits the best FID on Visual Genome and the second-best FID on MSCOCO as can be seen in the main paper Table 1. Also, although MAT_pretrained shows the best FID on MSCOCO, it was pretrained on the Places dataset [1] 80 times larger than MSCOCO, while Refill was trained on MSCOCO only. Despite having been trained on a (way) smaller dataset, Refill shows a comparable FID number: 7.192 (Refill) vs. 7.284 (MAT_pre). Such results show Refill’s superiority in producing intended instances accurately and realistically.
>
> [1] Zhou, B., Lapedriza, A., Khosla, A., Oliva, A., & Torralba, A. (2017). Places: A 10 million image database for scene recognition. IEEE transactions on pattern analysis and machine intelligence, 40(6), 1452-1464.
>
> > [“No Instance” and “Instance Across Missing Region”]
>
> > Q.3.3. It will be more convincing to provide methods to determine whether there are objects in the missing ("no instance"), and how to deal with cases that half of an object is missing while the other half is not ("instance across missing region").
>
>  We inform that **Refill can handle the case where the missing region may not include a visual instance, as shown in Appendix Figure 10 and Table 3**. Although predicting whether an instance should exist in a missing region is a subjective problem (If there is a missing region on a field of grass, should the model fill in the region with an object, like a dog or just grass?), Refill can fill in the missing region with no object if directed to do so by simply skipping the foreground object prediction. We provide the results in **Appendix Figure 10**, where the foreground object prediction is discarded, and only the background prediction is used. Furthermore, we provide the FID score of the results in **Appendix Table 3**. The FID scores in COCO-panoptic and Visual Genome are comparable to the other baselines (vanilla Refill, MAT, and HVITA), which demonstrates the effectiveness of Refill in “No instance” cases too.
>
>  Although we mentioned the “awareness of instance absence” problem in the Sec.5 limitation part of the main paper, we did not explicitly elaborate on a solution for it in the main paper. However, one promising approach would be to apply a filtering threshold at the end of the missing instance inference transformer. If no class probability achieves beyond the threshold, it is likely to be a case where just an empty background makes more sense. With this filtering-threshold method for “no instance” detection and the modified pipeline mentioned here, we believe object removal or filling in with “no instance” can be automated as well.
>
>  Also, we show that Refill can complete the image in the case that half of an object is missing while the other half is not. For example, in the left image of **Figure 5**, the part of the wood table and an entire plate are masked out. Refill is able to generate a visually convincing cup and missing part of the wood table.
>
>  We politely inform you that we have added **Section A.2, Figure 10, and Table 3 in the Appendix** during the revision period to address this concern.

---

> ### Author Response · Authors · 2022-11-25
> **Response to Reviewer 675z (2/3)**
>
> > [DETR prediction result after inpainting]
>
> > Q.4. It will be interesting to show whether the predicted instance in the impainted image can be detected with DETR. If yes, is the predicted category the same as the output of the "Missing Instance Inference Transformer"?
>
> We conduct additional experiments to address this question. In the Table below, **we present the percentage of cases where DETR’s predicted class is equal to the class of the “Missing Instance Inference Transformer” output after completing the images**. We also measure the DETR upper-bound performance using the target instances in original images as shown in the below Table (65.37/70.60% in COCO-panoptic and Visual Genome). Refill reaches 82.90% and 79.25% of the upper bound performance. Refill, thus, generates visual instances plausible enough for DETR to classify as the class used to generate instances.
>
> ||COCO-panoptic|Visual Genome(Zero-shot)|
> |:---:|:---:|:---:|
> |Refill|54.20%|55.95%|
> |Original|65.38%|70.60%|
>
> > [Academic Novelty]
>
> We would like to clarify our novelty as follows:
>
> First, we introduce two evaluation metrics CLIPScore and Visual Grounding Accuracy in the image inpainting scenario, where the visual instance is entirely removed out. The only prior work, HVITA, handles the same scenario with the lack of proper metrics to check the validity of generated visual instances.
>
> Second, we present the effective missing instance inference transformer to predict the missing instance in a scene. The module equipped with carefully designed positional encoding variants, especially ABS4C, could functionally produce the missing instance’s class by considering both semantically related surroundings and its location. We further demonstrate its effectiveness and generalizability by conducting additional experiments in Appendix A.1 of the paper.
>
> Third, Refill overcomes major limitations of HVITA : (1) exhibition of abrupt change along the boundaries, (2) lack of ability to handle various masks, and (3) heavy reliance on a refinement network.
>
> Fourth, The transformer-body background segmentation completion network shows better-recovered segmentation masks, especially under the presence of large missing regions.
>
> Fifth, Refill can hallucinate scene/context-friendly visual instances while harmonizing well with the unmasked areas. Refill outperforms baselines based on two introduced metrics (CLIPScore and Visual Grounding Accuracy) and exhibits comparable FID performance compared to SOTA approaches.
>
> Sixth, Refill can complete images with two versions (both “no instance” and “generate scene/context-friendly instance”), which previous works have no control over these options. Please see the above **[“No Instance” and “Instance Across Missing Region”]** reply for further details.
>
> > [Limited to DETR]
>
> > Q.1. Heavily limited to the pretrained detector. The proposed pipeline relies on DETR to provide the objects in the image, subsequent modules is limited to the pre-defined object categories of DETR. In the experiment design, only supported objects are masked as missing. Such heavy dependency limits the real-world applications of the method.
>
> Although our model (Refill) relies on DETR for objection detection and panoptic segmentation, the DETR part of Refill can be replaced with any other off-the-shelf panoptic segmentation model, such as Mask2Former [1]. This is because the missing instance inference transformer of Refill gets only bounding box coordinates and visual instances’ class and does not require finetuning the DETR model for missing instance prediction.
> The above argument states that our model can be easily extended to instance inpainting for wider categories, but we acknowledge reviewer 675z’s first comment that Refill is limited to the predefined object categories on which the prior module was trained. Considering object detection and panoptic segmentation models are also trained on specific categories, we insist that the limitation of Refill (limited to the pre-defined object categories) can be resolved by training the model using a more general panoptic segmentation model with a larger dataset in the future.
>
> [1] Cheng et al., Mask2Former: Masked-attention Mask Transformer for Universal Image Segmentation, CVPR 2022

---

> ### Author Response · Authors · 2022-11-25
> **Response to Reviewer 675z (1/3)**
>
> We thank reviewer 675z for the constructive comments. The reviewer appreciates the effectiveness of Refill over HVITA but pointed out the lack of novelty and the case of “no instance and instance across the missing region” along with other concerns. Here, we try to answer the reviewer's concerns one by one.
>
> > [Image Completion Scenario]
>
> > Q.2. The missing region is within or around the instance bbox, and the method predicts one object regardless of the existence or the number of objects. In many inpainting scenario, the missing region doesn't does not necessarily contain an object in the middle of the region.
>
> > Q3.1. Experimental design is biased. The datasets are prepared in favor of the proposed method, i.e., COCO and Visual Genome contains many objects; The missing regions are created around a carefully selected subset of objects.
>
> > Q.3.4. Meanwhile, experiments should be conducted on the full (not selected by object category) COCO-panoptic and Visual Genome dataset, with random missing region instead of around object bbox.
>
> Most conventional image inpainting methods address the restoring image when the parts of an instance are masked out. In particular, they have focused on center-aligned datasets (such as FFHQ and CelebA) in which parts of the face are missing. Also, they have shown  pluralistic generation in background-dominant scenes such as buildings and natural photography (Paris Street View and Places 2). In order to restore these images, instance-level information from a recognition model is not necessarily needed. Feature-level information from a stack of learning-based models is sufficient to restore the parts of the instance or background.
>
> In this paper, we cope with a hardly addressed image inpainting scenario, where the entire visual instance is removed. Only using feature-level information leads the model to lack the ability to generate scene/context-friendly instances in missing regions. To this end, we conclude to leverage instance-level context from DETR, similar to HVITA, which is the only prior work of this image inpainting scenario. Moreover, to learn ample amounts of inter-instance relations between multiple instances in the scene and ensure the removal/restoration of various types of missing instances, we use COCO-panoptic and Visual Genome datasets.
>
> In addition, we would like to mention that traditional image inpainting works also starts from the simple problem (rectangular mask in the center of the image) and extends into challenging setting (free-from mask such as scribble). Moreover, the methods to handle large hole sizes draw attention these days, where the hole size was small at the earlier time. As image inpainting technology evolves, we hope that the image inpainting scenario we dealt with will be extended into more general scenarios (multiple instances in a single missing region with various scales/locations and not restricted missing regions around/within instances’ bbox). Please note that Refill can handle various shapes of masks, while HVITA mainly targets rectangle masks and thus lacks generalization to other mask forms.
>
> Yet, we could not use full classes of COCO-panoptic things classes, we still use 10 classes more than HVITA settings. (HVITA uses 20 classes. Refill consider 30 classes). **Moreover, Refill is open to more than just the generation of only context-friendly instances. Refill is able to perform “object removal”, which is widely studied in the field of image completion.** Further details about "object removal" can be found in **[“No Instance” and “Instance Across Missing Region”]** reply.
>
> Please note that we change the title of the paper “Context-Aware Image Completion” to “Instance-Aware Image Completion” to emphasize the difference between conventional image inpainting tasks and our project.

---

> ### Author Response · Authors · 2022-12-11
> **Hope to continue further discussion with Reviewer 675z**
>
> First of all, thank you once more for your insightful thoughts.
>
> We attentively figure out your concerns and have made an effort to address them, particularly those related to the "academic novelty", "DETR dependency", "DETR prediction result after completion", "no instance and instance across missing region problem” and "our problem setting".
>
> We are looking forward to hearing back from you for the further discussion. Thanks.

---

### Author Response · Authors · 2022-11-25
**Overall Responses to All Reviewers**

We would like to express our best gratitude to all of the reviewers for their in-depth and meaningful comments. As a general comment, we summarize frequently discussed issues and address some of the key changes made accordingly. We look forward to receiving additional feedback and constructive discussion afterward. Detailed one-by-one replies can be found in each review’s comment. The revised parts are denoted with blue text in the manuscript.

Key changes during the revision period are briefly listed below:

> * We change the title of the paper “Context-Aware Image Completion” to “Instance-Aware Image Completion” to emphasize the difference between conventional image inpainting tasks and our project. [R1, R2, R3]
> * We report the performance (FID score) with the “no instance”(object removed) version of Refill and add some qualitative results in Table 3 and Figure 10 of the Appendix. Further details can be found in Appendix A.2. [R1, R3]
> * We experimentally measure the percentage of cases where DETR’s predicted class is equal to the output class of the “Missing Instance Inference Transformer” after completing the images. [R1]
> * We clarify the problem setting of our project and elaborate on the importance and the necessity of the recognition model (DETR). [R1, R3]
> * We analyze the color shifting issue. [R2, R3]
> * We elucidate the reason for the worse FID and LPIPS. [R2]
> * We report the classification accuracy of the “Missing Instance Inference Transformer” using the Visual Genome dataset (Unseen) in Table 2 of the Appendix to check the generalization ability. [R2]
> * We visualize the intermediate self-attention score of the “Missing Instance Inference Transformer” to investigate the dependency between surrounding instances and the predicted (missing) instance in Figure 8 of the Appendix. [R2]
> * We visualize how the predicted class from the “Missing Instance Inference Transformer” changes over different missing regions in Figure 9 of the Appendix. [R2]

---

### Author Response · Authors · 2022-12-07
**Regarding further questions or concerns.**

Thank you again for your insightful comments.

Please let me know if you have any questions about our responses.

We look forward to your further questions and discussions.

---

### Decision · Program_Chairs · 2023-01-20

**Decision:**

Reject

**Justification For Why Not Higher Score:**

This paper receives 2x marginally below the acceptance threshold and 1x marginally above the acceptance threshold.

**Justification For Why Not Lower Score:**

NA

**Metareview: Summary, Strengths And Weaknesses:**

This paper receives 2x marginally below the acceptance threshold and 1x marginally above the acceptance threshold. The overall rating is leaning towards rejection. The major weakness of the paper is the lack of academic novelty where paper provides an incremental improvement over HVITA, but does not solve critical problems like "no instance" or "instance across missing region". The model performance is highly dependent on the missing instance prediction module. Nonetheless, there is a lack of sufficient explanation and analysis to prove the effectiveness, generalization, and robustness of this module.

The metareviewer carefully reads the authors' response to the lack of novelty over HVITA and finds that besides introducing new evaluation metrics and claiming the Refill model outperforms HVITA, there is no detailed explanation of what are the technical differences between HVITA and the proposed model. Although there is a 1x marginally above the acceptance threshold, the strengths highlighted by the reviewer do not address the novelty claim of the proposed method over HVITA.